# Shallow-angle needle guide for ultrasound-guided internal jugular venous catheterization: A randomized controlled crossover simulation study (CONSORT)

**Kunitaro Watanabe**[1][☉]*, **Joho Tokumine**[1][☉], **Alan Kawarai Lefor**[2][‡], **Tomoko Yorozu**[1][‡]

**1** Department of Anesthesiology, Kyorin University School of Medicine, Mitaka, Tokyo, Japan, **2** Department of Surgery, Jichi Medical University, Shimotsuke, Tochigi, Japan

☉ These authors contributed equally to this work.
‡ These authors also contributed equally to this work.
* kunitarowatanabe@yahoo.co.jp

**Data Availability Statement:** All relevant data are within the manuscript and its Supporting Information files.

## Abstract

### Background

Needle guides for ultrasound-guided internal jugular venous catheterization facilitate successful cannulation. The ability of a needle guide to prevent a posterior vein wall injury which may secondarily induce lethal complications, is unknown. Previous studies showed that a shallow angle of approach may reduce the incidence of posterior wall injuries. We developed a novel needle guide with a shallow angle of approach for ultrasound-guided venous catheterization and examined whether this needle guide reduces the incidence of posterior wall injuries compared to a conventional needle guide and free-hand placement in a simulated vein.

### Methods

This study was a randomized crossover-controlled trial. The primary outcome was the rate of posterior vein wall injuries. Participants had a didactic lecture about three ultrasound-guided techniques using the short-axis out-of-plane approach, including free-hand (P-free), a commercial needle guide (P-com), and a novel needle guide (P-sha). The view inside a simulated vein was recorded during venipuncture.

### Results

Thirty-five residents participated in this study. Posterior vein wall injuries occurred in 66% using P-free, 60% using P-com, and 0% using P-sha (p< 0.01). There was no significant difference in the incidence of posterior vein wall injuries between P-free and P-com.

**Funding:** The authors received no specific funding for this work.

**Competing interests:** The authors have declared that no competing interests exist.

## Conclusions

Use of a shallow angle of approach needle guide resulted in a lower rate of posterior vein injuries during venipuncture of a simulated vein compared with other techniques using a steeper angle techniques.

## Background

Ultrasound guidance during central venous catheterization is associated with high success rates and low mechanical complication rates, and has been recognized as the "gold standard" technique [1]. Sufficient training is required to perform successful ultrasound-guided venous catheterization, which includes an emphasis on needle visualization, hand-eye coordination and avoiding posterior vein wall injuries. However, the nature of optimal training is unclear. A recent report showed that mechanical complications during ultrasound-guided internal jugular venous catheterization have a 4% incidence [1].

Needle guides have been developed to assist operators during ultrasound guided venous catheterization. Popular needle guides for ultrasound-guided central venous catheterization are classified as mechanical needle guides, in which the needle trajectory is mechanically determined by the guide while advancing toward the target vein. Mechanical needle guides for the in-plane approach allow clear visualization of the entire needle, from the tip to the shaft of the needle [2]. Mechanical needle guides for the out-of-plane approach allow good visualization of the needle leading to precise depth in the center of the ultrasound view [3]. Needle guides may increase the success rate of ultrasound-guided central venous catheterization. However, the rate of arterial injuries was reported to be almost the same compared to the free hand technique of ultrasound-guided central venous catheterization [4].

A previous study showed that a shallow angle of approach of the needle may reduce the incidence of posterior vein wall injuries [5]. We developed a new needle guide to assure a shallow angle of approach for internal jugular venous catheterization. In this study, we evaluate the success rate and rate of posterior vein wall injuries using the newly developed needle guide, a conventional commercial needle guide, and the free hand method using a simulated internal jugular vein.

## Materials and methods

This study was reviewed and approved by the local ethics committee (Kyorin University Ethical Review Board, Reception No. H30-022) and registered in the University Hospital Medical Information Network Center Clinical Trials Registration System (UMIN000030151, 2018/6/7). Participants were recruited from among first-year residents as volunteers (Fig 1). Written informed consent was obtained from all participants. Participant recruitment and data collection were performed from June 2018 to February 2019. Potential participants with prior experience performing central venous catheterization with using a needle guide were excluded. Sample size was calculated using a previous study [5] which is described below in the statistical analysis section.

### Needle guide

The newly developed needle guide was designed to result in a shallow angle of approach using the short-axis out-of-plane method for internal jugular vein catheterization and created using

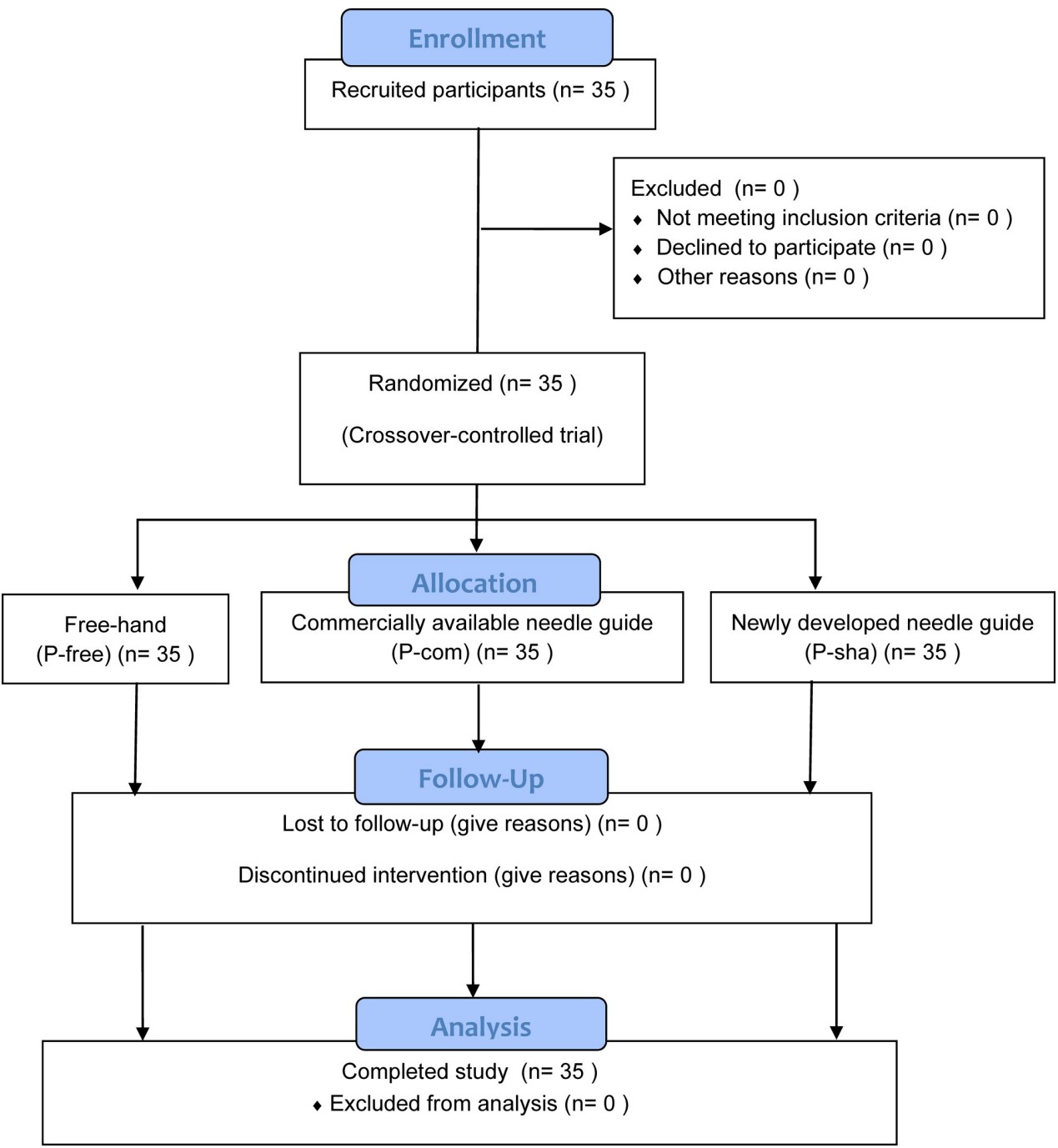

**Fig 1. Consort flow chart.**

the 3D modeling software 3D Builder (Microsoft Co., USA). The needle guide was made of nylon using a 3D print service (DMM.make Co., Japan) (Fig 2). The angle of approach to the target vein was set at 28˚, which is shallower than the usual angle of approach of 60˚ with a conventional commercial needle guide. The depth of the needle tip on the ultrasound view was designed to be 1.5 cm and initial skin insertion site was 2.5 cm from the ultrasound probe. In this study we defined a shallow angle as under 30˚. The commercial needle guide used for the

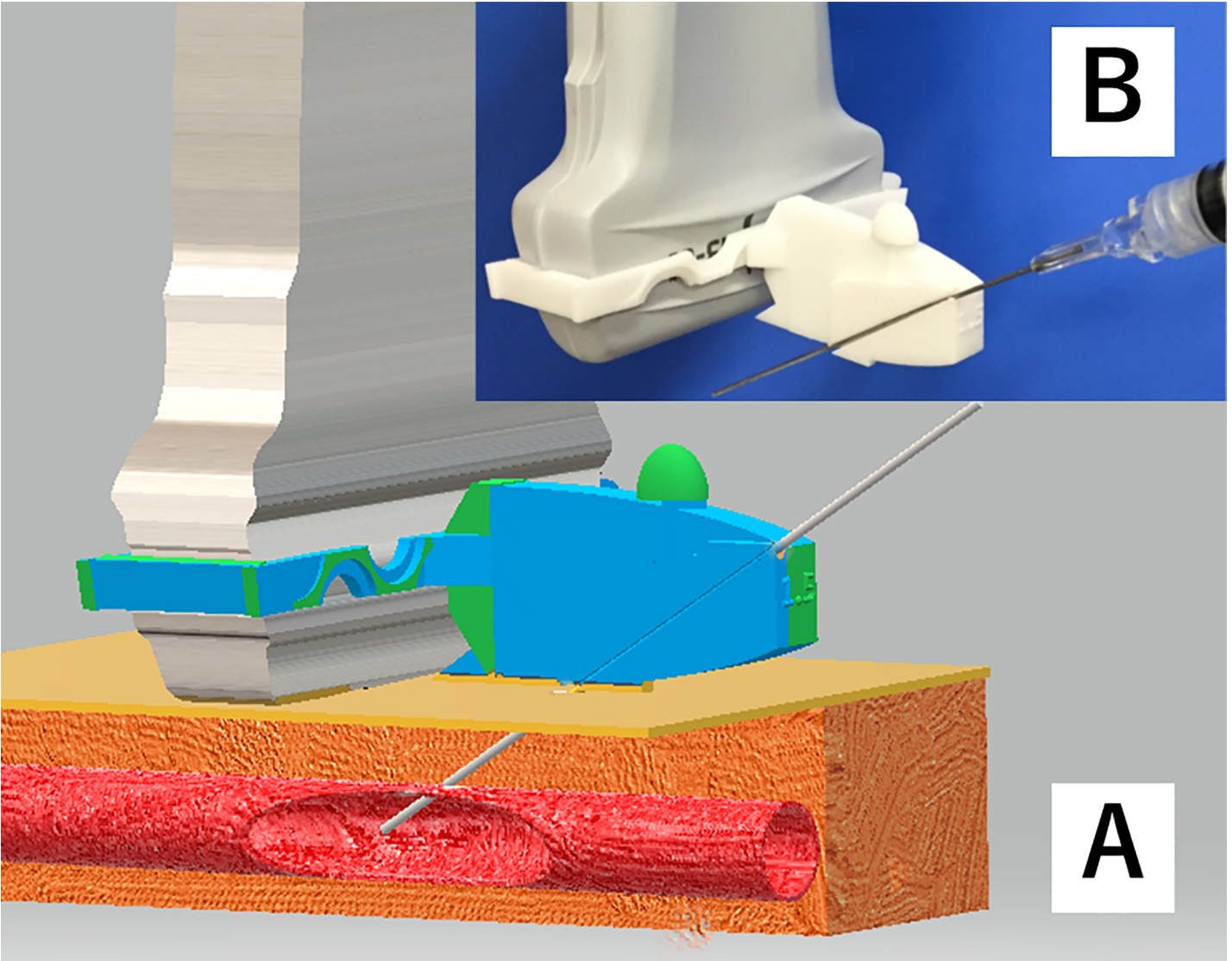

**Fig 2. Novel needle guide with a shallow angle of approach.** Panel A: Design of the novel needle guide using 3D Builder. Panel B: The needle guide is created using a 3D printing service and is made of nylon.

short-axis out-of-plane approach was the AccuSITE™ (Civco Co., USA), which can be set to a depth of the needle tip at 1.5 cm.

## Ultrasound, needle, and vein simulator

The ultrasound machine used was the EDGE HFL38 (FUJIFILM SonoSite, Inc., USA). The central venous catheterization needle was a 20 G metal introducer needle CV Legaforce EX® (effective length 50 mm, Termo Co., Japan) [6]. The vein simulator was the UGP GEL® (ALFABIO Co., Japan), in which a simulated internal jugular vein and carotid artery are located 11 mm and 22 mm under the surface. The simulated internal jugular vein was connected to a water tank through a tube to maintain pressure at 10 cm $H_2O$, which was monitored by a pressure transducer. An endoscope was inserted into the simulated internal jugular vein and an inside view of the vessel recorded during the procedure.

### Simulation study

This study was planned as a randomized crossover-controlled trial. Participants received a didactic lecture to learn basic skills in ultrasound-guided central venous catherization. Instructors demonstrated how to perform three approaches, including free-hand (P-free) (Fig 3), using a commercially available needle guide (P-com) (Fig 4), and using the newly developed needle guide (P-sha) (Fig 5). The technique for the free hand procedure was presented according to the guidelines of the American Society of Echocardiography and the Society of Cardio-vascular Anesthesiologists [7].

During simulation training, participants could see the inside of the simulated vessel using an endoscope incorporated into the simulator. The goal of the hands-on training was successful venipuncture of the simulated vein without causing a posterior vein wall injury. Individualized one-to-one guidance was provided during the two-hour hands-on training session to assure acquisition of proper skills.

After simulation training, all participants performed each of the three approaches using the simulator and their performance was evaluated. The endoscopic view inside the simulated vessel could not be seen by participants but was recorded for later review. The video recordings were sequentially numbered, but this number was later randomized by computer to maintain

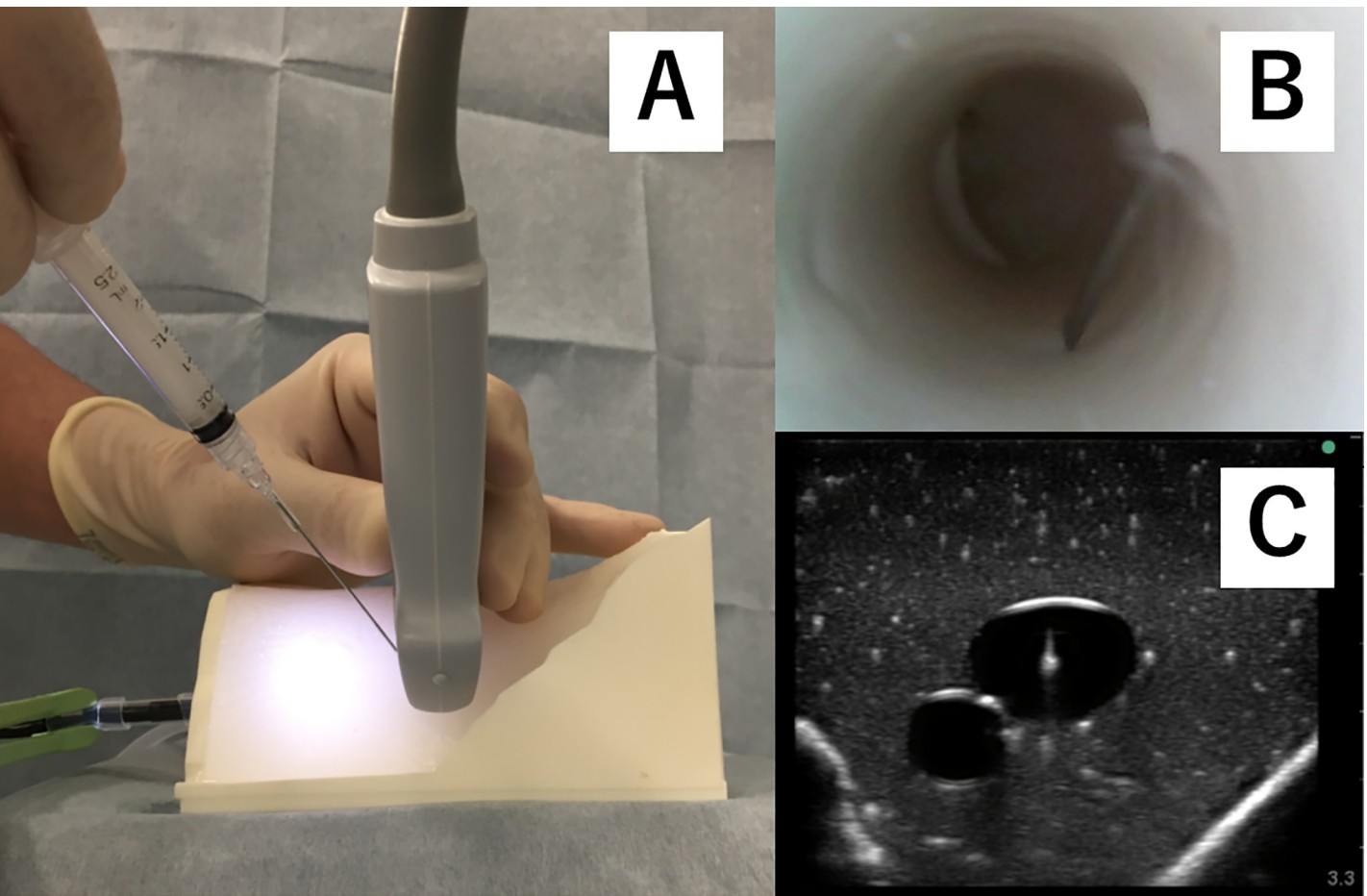

**Fig 3. Free-hand short-axis out-of-plane approach.** Panel A: Needle insertion starts close to the ultrasound probe, and the angle of approach is relatively steep, according to the guidelines of the American Society of Echocardiography and the Society of Cardiovascular Anesthesiologists. Panel B: Internal view of the simulated vein. Panel C: Ultrasound-view of the simulated vein.

anonymity. The sequence for each participant performing each technique was randomly decided using a random number table. The blinding of data and study group allocations were performed by a person who did not participate in the trial. The technique used and individual identification were concealed for the evaluation. Two senior physicians who did not participate in the test, observed the recorded videos and evaluated whether the procedure was performed successfully or not.

The primary outcome of this study was the rate of posterior vein wall injuries. A posterior vein wall injury was defined as obvious penetration of the posterior vein wall by the needle on visual inspection. Secondary outcomes included success rate, number of needle passes till success, time for the procedure, and unanticipated arterial injuries. A questionnaire was given to participants to evaluate comfort and the preferred procedure using a 5-point Likert scale (5: very comfortable, 1: uncomfortable).

## Statistical analysis

**Sample size calculation.** A previous study showed that the incidence of posterior vin wall injury is 41% using a steep angle of approach and 9% with a shallow angle of approach [5]. Based on this data the sample size required for 80% power at ɑ = 0.05 was estimated to be thirty-four participants and we planned to include thirty-five junior residents as participants. Sample size was calculated with EZR using R commander, version 1.32 (Saitama Medical Center, Jichi Medical University, Saitama, Japan) [8].

Fisher's exact test was used to evaluate the rates of posterior vein wall puncture, arterial puncture, success, and the Likert-scale. Analysis of variance (ANOVA) and the Bonferroni correction were used to compare continuous variables. Numerical values were expressed as ratios (%) or as the mean ± standard deviation for normal distributions, and as the median [interquartile range] for non-normal distributions. A p-value less than 0.05 was considered statistically significant. Statistical analyses were performed with EZR using R commander, version 1.32 (Saitama Medical Center, Jichi Medical University, Saitama, Japan) [8].

## Results

Thirty-five residents participated in this study, with no exclusions. Only four participants had previous experience performing central venous catheterization (Table 1). No participant had prior experience performing central venous catherization using needle guide.

There were no posterior vein wall injuries in the P-shal (0%) group (newly developed guide), which was significantly lower than when using the other two techniques (p< 0.01) (Table 2). There was no significant difference between the P-free and P-com groups. The procedure time using P-free was significantly longer (p < .05) than the other two techniques, but there was no difference between P-com and P-shal (p>.05). There was no significant difference in the number of needles passes among the three techniques (p = 1.00). There were no arterial injuries in any procedures. All procedures had a 100% success rate.

The subjective comfort level for those in the P-free group was significantly less (higher score) than P-com and P-shal (p < 0.01). There were no significant differences between P-com and P-shal (p = 1.00). The most preferred procedure was P-com.

## Discussion

This study demonstrates that a novel needle guide with a shallow angle of approach for the short-axis out-of-plane approach of ultrasound-guided internal jugular venous catheterization resulted in significantly fewer posterior vein wall injuries in a simulated vein. Some studies have shown that that the conventional free-hand short-axis out-of-plane approach is associated

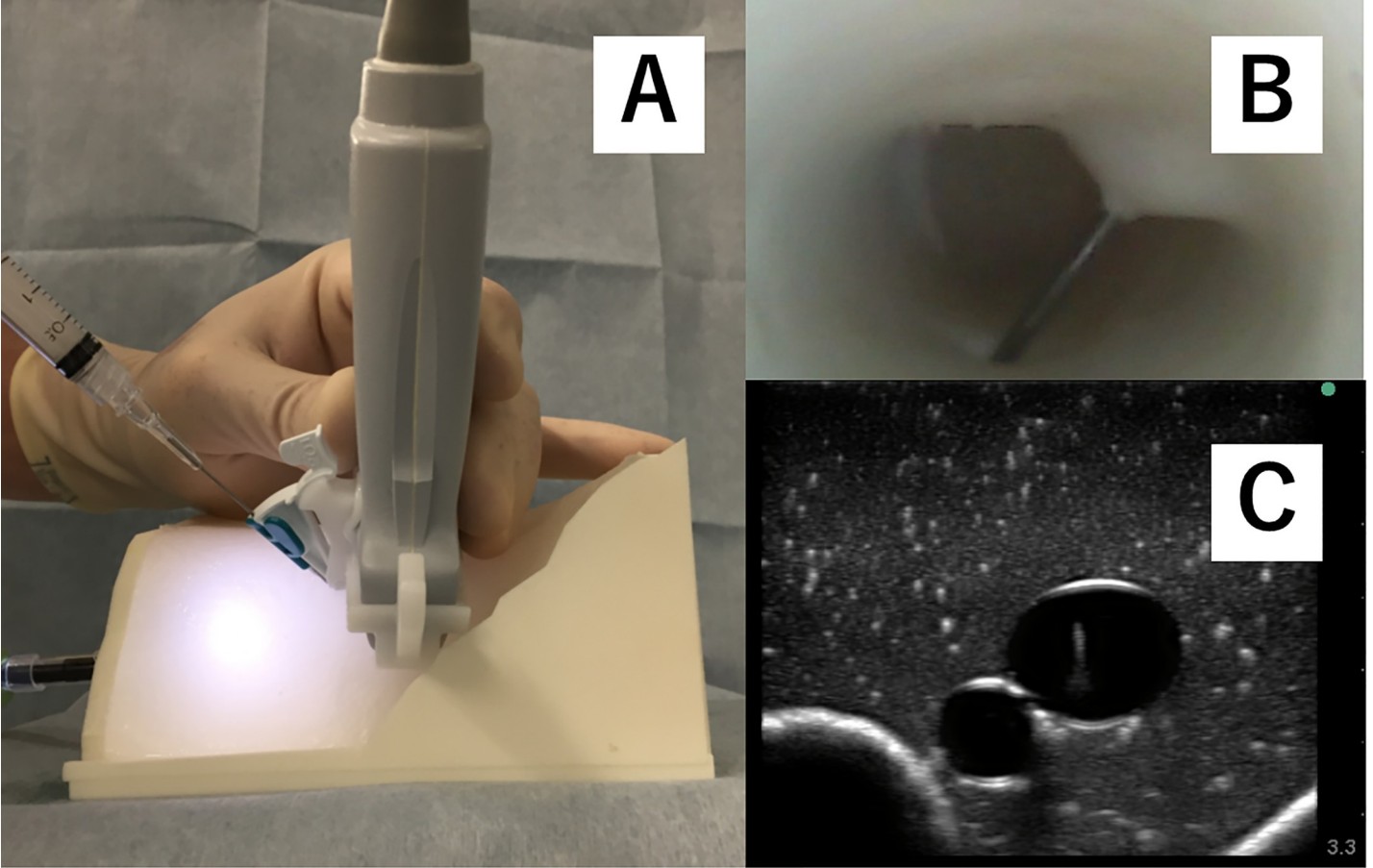

**Fig 4. A commercial needle guide for out-of-plane approach.** Panel A: The needle is inclined at 60 degrees to the target vein. Panel B: Internal view of the simulated vein. Panel C: The needle tip appears in the ultrasound image at a depth of 1.5 cm.

with a high risk of posterior wall injury [5,9]. A previous study showed that the risk of posterior vein wall injury is not due to the short-axis out-of-plane approach itself, but rather due to the steeper angle of approach [5]. If the angle of approach is changed from 60° to 28°, the calculated needle trajectory path from the anterior vein wall to the posterior vein wall will be approximately 1.7 times longer. This may partially explain shallow angle of approach reduces the rate of posterior vein wall injuries. Expert operators performing ultrasound-guided central venous catheterization may stop the needle tip before reaching the posterior vein wall, even with a steep angle of approach. However, those with less experience may not be able to stop advancing the needle when using a steep angle of approach. We suggest that a shallow angle of approach may limit advancing the needle tip when less experienced operators perform the short-axis out-of-plane approach.

The long-axis in-plane approach for ultrasound-guided internal jugular venous catheterization may be recommended to reduce the risk of posterior vein wall injury, because the operator can observe both the needle tip and the posterior vein wall during the procedure [10]. The guideline of the American Society of Echocardiography and the Society of Cardiovascular Anesthesiologists recommends using the short-axis out-of-plane approach for ultrasound-guided internal jugular venous catheterization as the standard approach [7]. The long-axis in-plane approach is difficult to use for internal jugular vein catheterization because of limited

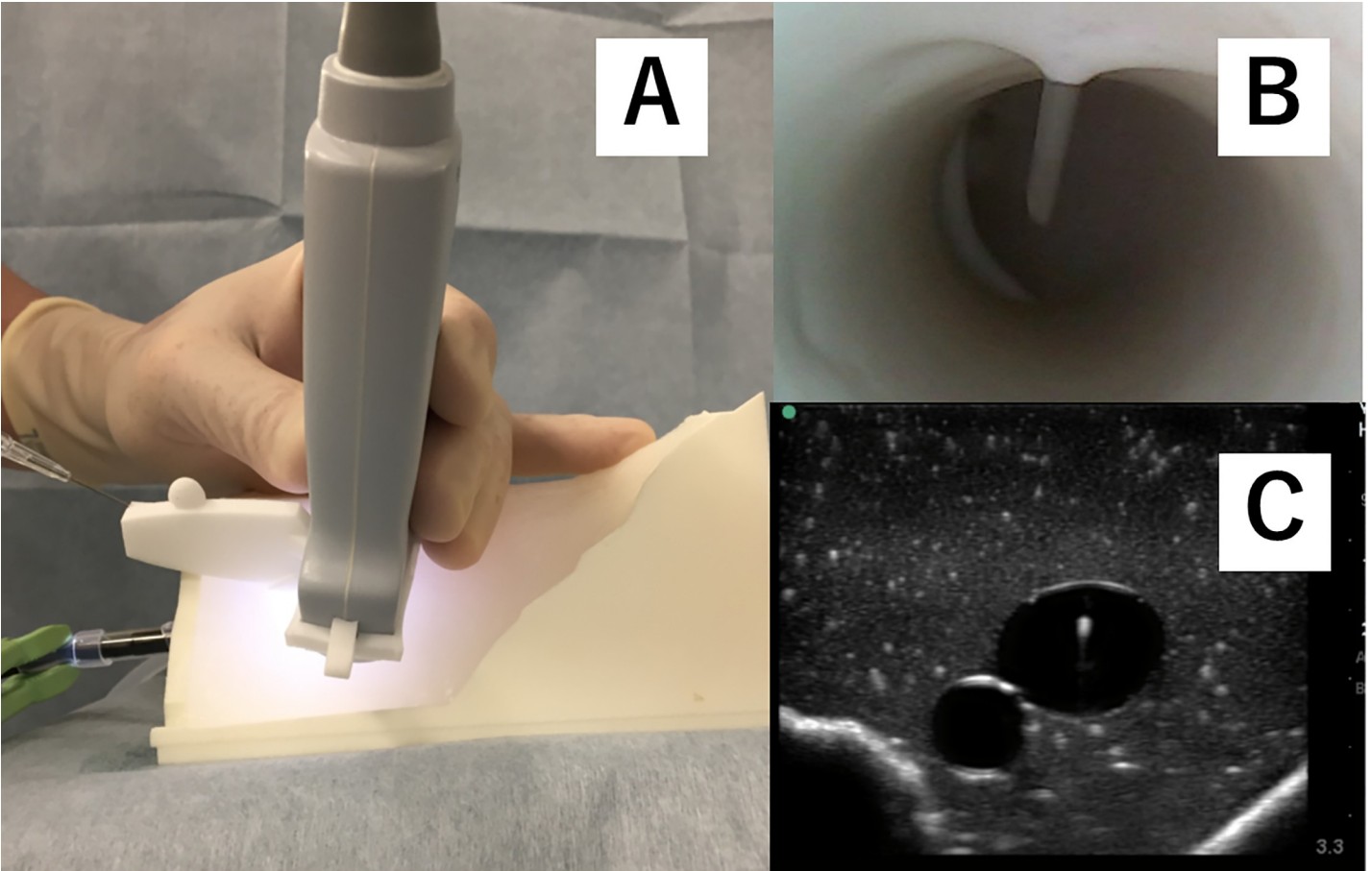

**Fig 5. A novel shallow angle of approach needle guide.** Panel A: The needle insertion site is far from the probe, 2.5cm cephalad and the needle is inclined at 30 degrees to the simulated vein, which allows a shallow angle of approach to the simulated internal jugular vein. Panel B: Internal view of the simulated vein. Panel C: The needle tip appears in the ultrasound image at a depth of 1.5 cm.

maneuverability of the ultrasound probe on the neck in patients with an ordinary physique. A small-footprint probe is needed to apply the long-axis in-plane approach. Furthermore, particular facility with real-time visualization and hand-eye coordination are needed to perform the long-axis in-plane approach in clinical practice.

The short-axis out-of-plane approach has the advantage of relating the insertion technique to the anatomical relationship of the internal jugular vein and the common carotid artery, which may reduce the incidence of unanticipated common carotid artery injuries. Oblique

**Table 1. Demographic data of participants.**

| | |
|---|---|
| **Prior months of residency training** | 5 ± 2 |
| **Gender (male: female)** | 23: 12 |
| **Age (years)** | 27 ± 2 |
| **Experience placing a CVC (yes: no)** | 4: 31 |
| **Experience- number of CVCs placed (none: 1: 2: 3)** | 31: 1: 2: 1 |
| **Experience with ultrasound-guided CVC placement (yes: no)** | 4: 31 |
| **Number of ultrasound-guided CVCs placed (none: 1: 2: 3)** | 31: 1: 2: 1 |

CVC: Central venous catheterization

**Table 2. Primary and secondary outcomes and questionnaire results.**

| Procedure | P-free | P-com | P-shal |
|---|---|---|---|
| Posterior vessel wall injury (%) | 66 | 60 | 0 |
| Arterial injury (%) | 0 | 0 | 0 |
| Overall success rate (%) | 100 | 100 | 100 |
| Needle passes till success (n) | 1.0 ± 0 | 1.0 ± 0 | 1.0 ± 0 |
| Time for procedure (seconds) | 32 ± 17 | 22 ± 9 | 23 ±10 |
| Comfort for procedure (5–1) (5: very comfortable, 1: uncomfortable) | 3[2,3] | 4[3,5] | 4 [3,5] |
| Preferred procedure (%) | 6 | 54 | 40 |

P-free: Free hand insertion, P-com: Commercial needle guide at 60$^\circ$, P-shal: a novel needle guide with a shallow angle of approach (28$^\circ$)

approaches were developed to overcome the disadvantages of the out-of-plane and in-plane approaches, while maintaining the advantages of both approaches [11, 12]. The oblique approach has the advantage of being directly related to the anatomical relationship between the vein and artery similar to the out-of-plane approach and preventing posterior vein wall injuries as with the in-plane approach. Unfortunately, the oblique approach has the same disadvantage as the in-plane approach, which is difficult needle handling. When using the oblique approach, the needle is directed toward the superior mediastinum, which increases the risk of an unanticipated arterial or venous injury in the superior mediastinum [11, 12]. When using the medial oblique approach, the needle is directed to the subclavian artery and apex of the lung [13].

In this study, only 4/35 participants had any experience and 31/35 were completely naïve to central venous catheterization. Hands-on simulation training for two hours resulted in a 100% success rate using the short-axis out-of-plane approach regardless of whether or not a needle guide was used. However, using a steeper angle of approach resulted in many posterior vein wall injuries despite using a needle guide. This study suggests that a steep angle of approach may be an important factor associated with posterior vein wall injuries.

This study was performed using a simulated vein. Therefore, the results may not predict the results of a clinical study. However, we believe that these results strongly support the idea that posterior vein wall injuries are not an inherent disadvantage of the out-of-plane approach, but are related to the steep angle of approach of the needle. Ultimately, we want to establish an ideal out-of-plane approach for ultrasound-guided internal jugular venous catheterization using a shallow angle of approach needle guide (The 3D data used to construct the shallow angle needle guide for the out-of-plane approach are supplied as a S1 File)

## Conclusions

A novel needle guide with a shallow angle of approach (28$^\circ$) for the out-of-plane technique resulted in a remarkably lower rate of posterior vein wall injuries in ultrasound-guided internal jugular venous catheterization performed by residents with little clinical experience compared with a commercial needle guide (60$^\circ$) or free hand technique in a study using a simulated vein. Further study of this novel shallow angle needle guide including clinical trials may be warranted based on these preliminary results.

## Supporting information

**S1 Checklist. CONSORT checklist.** CONSORT 2010 checklist of information to include when reporting a randomized trial.
(PDF)

**S1 File. 3D Data for constructing the novice shallow angle needle guide.**
(ZIP)

**S2 File. Study protocol.**
(PDF)

## Acknowledgments

The authors thank contribution for the study to junior residents of Kyorin University Hospital.

## Author Contributions

**Conceptualization:** Kunitaro Watanabe, Joho Tokumine.

**Data curation:** Kunitaro Watanabe, Joho Tokumine.

**Formal analysis:** Kunitaro Watanabe, Joho Tokumine.

**Investigation:** Kunitaro Watanabe, Joho Tokumine.

**Methodology:** Kunitaro Watanabe.

**Software:** Kunitaro Watanabe, Joho Tokumine.

**Supervision:** Alan Kawarai Lefor, Tomoko Yorozu.

**Validation:** Kunitaro Watanabe, Joho Tokumine.

**Visualization:** Alan Kawarai Lefor, Tomoko Yorozu.

**Writing – original draft:** Kunitaro Watanabe, Joho Tokumine.

**Writing – review & editing:** Alan Kawarai Lefor, Tomoko Yorozu.

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
