## [Decision Letter · Decision Letter 0]

12 Jun 2020

PONE-D-20-10483

Shallow-angle needle guide for ultrasound-guided internal jugular venous catheterization: A pilot simulation study

PLOS ONE

Dear Dr. Kunitaro Watanabe,

Thank you for submitting your manuscript to PLOS ONE. After careful consideration, we feel that it has merit but does not fully meet PLOS ONE’s publication criteria as it currently stands. Therefore, we invite you to submit a revised version of the manuscript that addresses the points raised during the review process.

We look forward to receiving your revised manuscript.

Kind regards,

Georg M. Schmölzer

Academic Editor

PLOS ONE

Journal Requirements:

'The funders had no role in study design, data collection and analysis, decision to publish, or preparation of the manuscript.'

4. Please ensure that you refer to Figure 5 in your text as, if accepted, production will need this reference to link the reader to the figure.

Additional Editor Comments (if provided):

Reviewers' comments:

Reviewer's Responses to Questions

**Comments to the Author**

1. Is the manuscript technically sound, and do the data support the conclusions?

Reviewer #1: Yes

2. Has the statistical analysis been performed appropriately and rigorously? 

Reviewer #1: Yes

3. Have the authors made all data underlying the findings in their manuscript fully available?

Reviewer #1: Yes

4. Is the manuscript presented in an intelligible fashion and written in standard English?

Reviewer #1: Yes

5. Review Comments to the Author

Reviewer #1: Interesting paper

Some issues.

1) posterior vein wall injuries should be defined

2) shallow should be better defined

3) the present is a randomized controlled trial; this should be better specified in the title

4) sample size calculation is missing

5) due to reduced sample size normal distribution should be tested for.

6. PLOS authors have the option to publish the peer review history of their article (what does this mean?). If published, this will include your full peer review and any attached files.

Reviewer #1: Yes: Fabrizio D'Ascenzo

---

## [Author Response · Author response to Decision Letter 0]

14 Jun 2020

Thank you for the opportunity to revise our manuscript. We have responded to the reviewer’s comments in a point-by-point fashion. Replies to the comments are shown in the revised manuscript in a red italic font. We believe that this has resulted in a strengthened manuscript.

 The authors received no specific funding for this work.

Thank you for your continued consideration.

---

## [Editor Report · Decision Letter 1]

17 Jun 2020

Shallow-angle needle guide for ultrasound-guided internal jugular venous catheterization: A randomized controlled crossover simulation study (CONSORT)

PONE-D-20-10483R1

Dear Dr. Kunitaro Watanabe,

We’re pleased to inform you that your manuscript has been judged scientifically suitable for publication and will be formally accepted for publication once it meets all outstanding technical requirements.

Kind regards,

Georg M. Schmölzer

Academic Editor

PLOS ONE
---

## [Editor Report · Acceptance letter]

19 Jun 2020

PONE-D-20-10483R1 

Shallow-angle needle guide for ultrasound-guided internal jugular venous catheterization: A randomized controlled crossover simulation study (CONSORT) 

Dear Dr. Watanabe:

I'm pleased to inform you that your manuscript has been deemed suitable for publication in PLOS ONE. Congratulations! Your manuscript is now with our production department. 

Kind regards, 

on behalf of

Dr. Georg M. Schmölzer 

Academic Editor

PLOS ONE